# DQ-Former: Querying Transformer with Dynamic Modality Priority for Cognitive-aligned Multimodal Emotion Recognition in Conversation

## ABSTRACT

Multimodal Emotion Recognition in Conversations aims to understand the human emotion of each utterance in a conversation from different types of data, such as speech and text. Previous works mainly focus on either complex unimodal feature extraction or sophisticated fusion techniques as general multimodal classification tasks do. However, they ignore the process of human perception, neglecting various levels of emotional features within each modality and disregarding the unique contributions of different modalities for emotion recognition. To address these issues, we propose a more cognitive-aligned multimodal fusion framework, namely DQ-Former. Specifically, DQ-Former utilizes a small set of learnable query tokens to collate and condense various granularities of emotion cues embedded at different layers of pre-trained unimodal models. Subsequently, it integrates these emotional features from different modalities with dynamic modality priorities at each intermediate fusion layer. This process enables explicit and effective fusion of different levels of information from diverse modalities. Extensive experiments on MELD and IEMOCAP datasets validate the effectiveness of DQ-Former. Our results show that the proposed method achieves a robust and interpretable multimodal representation for emotion recognition.

## CCS CONCEPTS

• **Information systems** → **Multimedia information systems**; **Sentiment analysis**; • **Computing methodologies** → **Discourse, dialogue and pragmatics**; • **Computer systems organization** → *Neural networks*.

## KEYWORDS

multi-modal fusion; middle-fusion; multi-modal emotion recognition in conversation

## 1 INTRODUCTION

Multimodal Emotion Recognition in Conversations (MERC) plays a crucial role in improving human-computer interaction by enabling machines to accurately discern emotions and respond accordingly. This task aims to understand human emotions conveyed in each utterance by leveraging diverse modalities like acoustic and textual

**Unpublished working draft. Not for distribution.**

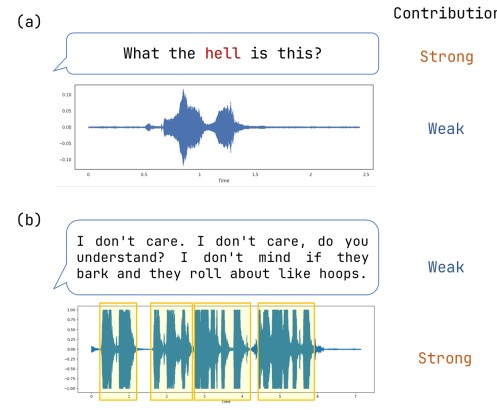

**Figure 1: Two examples from IEMOCAP. Not all modalities contribute equally to emotion recognition. In case (a), emotions can be readily inferred from the text, whereas weak pitch in the voice, conversely, tends to introduce disruptive noise. In case (b), a strongly intense tone of voice can clearly express the speaker's anger, whereas the long context in textual modality can be challenging to understand the emotion.**

cues [24]. The efficacy of MERC models heavily hinges on the quality of multimodal representations.

Previous works mainly treat MERC as a general multimodal classification task. The dominant framework involves unimodal feature extraction, information interaction, and multimodal fusion. Some investigations underscore the pivotal role of high-quality unimodal features, emphasizing their direct impact on multimodal representations [28, 41]. Certain works delve into modeling inter-modal or intra-modal interactions [23, 38]. Additionally, researchers develop sophisticated strategies for multimodal fusion [8, 31, 37].

However, such generalized frameworks tends to ignore the intricacies of human emotion expression and perception. Humans express emotions not only through semantics, but also through nuanced modality-specific attributes, such as pitch, loudness, and tempo in their voice [6, 13]. Existing late-fusion works, which rely on the final representations of unimodal models for multimodal fusion, fail to leverage the unique attributes of each modality [35]. Besides, the contribution of each modality vary under different circumstances for emotion recognition [7]. As illustrated in Figure 1, while text emerges as the dominant modality for clearly conveying anger in one instance (case a), the loud voice signal could serve as the primary indicator of emotion in another (case b). Disregarding the relative importance of different modalities and fine-grained modality-specific attributes during multimodal fusion may result in suboptimal performance of MERC systems [30].

Recent studies indicate that pre-trained unimodal models contain rich hierarchical information across various layers [12, 18]. For example, interpretability studies in speech models [3, 18, 26] show that paralinguistic and linguistic properties are encoded in different layers. This phenomenon is mirrored in text models as well [12]. Generally, the final representations of unimodal models are mainly associated with semantic-level information, while various levels of modality-specific information are distributed across intermediate layers [3, 12]. These findings indicate the potential for the mid-layer fusion strategy to capture diverse levels of emotional cues within advanced modality-specific pre-trained models [35].

Motivated by the above observations, we propose a more cognitive-aligned middle-fusion multimodal framework, namely DQ-Former, to enhance the modeling of emotional cues for a robust MERC system. Specifically, we first encode each modality using modality-specific pre-trained models. Then we employ a small set of learnable queries to collate and condense emotion cues embedded in intermediate layers of each unimodal encoder through layer-wise cross-modal attention. Finally, we aggregate various granularities of emotion cues based on their confidence in emotion recognition. In this sense, the cognitive-aligned relationships between multiple modalities are explicitly reasoned, achieving a complementary and robust multimodal representation for emotion recognition.

Our main contributions can be summarized as follows:

- We present DQ-Former, a cognitive-aligned middle-fusion framework that leverages various granularities of emotion cues from each modality, since human emotions are conveyed not only through semantics but also through specific attributes.
- We devise a novel dynamic modality priority learning module to aggregate multiple modalities, which explicitly manages the contribution of each modality for every instance.
- DQ-Former achieves impressive performance on MELD and IEMOCAP datasets. Our analysis reveals that the proposed method establishes a robust and interpretable multimodal representation for emotion recognition.

## 2 RELATED WORK

### 2.1 Multimodal Fusion Works

The self-attention mechanism introduced in the transformer architecture [32] offers a versatile approach for modeling signals across diverse modalities. However, recent advanced models tend to specialize in particular modalities and are tailored for unimodal tasks [1, 3, 4, 20]. Hence, the dominant framework of previous MERC systems adopts a two-phase late-fusion pipeline: *first* extracting feature representations of each modality separately simply utilizing the final representations from pre-trained unimodal models [28, 41]; and *then* employing sophisticated fusion techniques to integrate them [22, 29, 37, 38]. A typical late-fusion framework is MulT [31], which uses directional pairwise cross-attention to align different modalities. MISA [8] projects each modality into a modality-specific subspace to capture their unique features and into a modality-invariant subspace to capture their shared characteristics. Sun et al. [29] employ unimodal transformer modules to model the representations of each modality and a multimodal

transformer module to fuse all the modalities. These works fail to consider fine-grained modality-specific attributes.

In addition to the two-phase late-fusion approach, there are efforts towards developing end-to-end early-fusion or middle-fusion MERC systems. One prevalent strategy involves directly combining different modalities into unified inputs for the transformer model [17]. This operation often ignores the heterogeneity among different modalities. On the other hand, Xu et al. [36] introduce a bridge-layer between the top layers of uni-modal encoders and each layer of the cross-modal encoder, so as to get a good vision-language representation with diverse levels of information. In addition, Nagrani et al. [25] propose a middle-fusion strategy MBT, which directs information exchange between modalities through bottleneck latent spaces, and achieves impressive performance on audio-visual benchmarks. Inspired by MBT, Wu et al. [35] introduce the multimodal recurrent intermediate-layer aggregation model for MERC task. Despite these advancements, all these works typically treat MERC as general multimodal tasks do. They lack consideration of different modality priorities for emotion recognition before fusion.

### 2.2 Modalities Priority Learning

Recent studies have shown that different modalities contribute uniquely to emotion recognition [7]. Existing works either rely on manually selected dominant modalities based on prior knowledge, or fail to fully exploit the dominant modalities and filter out the misleading signal. For example, Zhang et al. [39] propose a language-guided multimodal fusion strategy as they believe the language modality usually stands out as the dominant one among all modalities. Wang et al. [33] utilize modality-specific softmax functions to compute the attention weights in a single-stream Transformer. Hong et al. [9] propose a model-agnostic auxiliary network to select the fusion modality. Wang et al. [34] propose a new architecture that uses dynamic modality gating to determine a primary modality and hierarchically incorporates other auxiliary modalities. Although these methods consider the varying contributions of different modalities, they often treat multimodal inputs generically, ignoring the distinctive properties of each modality.

Different from all these works, we propose a new middle-fusion framework. It not only considers various levels of emotion cues across multiple intermediate layers, but also fuses them with a dynamic modality priority.

## 3 PROPOSED FRAMEWORK

In this section, we introduce a more cognitive-aligned multimodal middle-fusion framework, namely DQ-Former. We first state the problem. Following that, we provide details of the unimodal encoder, the proposed multimodal fusion framework, and dynamic modality priority learning. The overall architecture of DQ-Former is depicted in Figure 2.

### 3.1 Problem Definition and Notation

Let $C = \{C_i\}_{i=1}^{N}$ denote a conversational dataset, where $N$ is the total number of conversations. Each conversation $C_i$ consists of $N^{(i)}$ utterances. Let $u_{ij}$ represents the $j$-th utterance in the $i$-th conversation $C_i$, where $j \in [0, N^{(i)})$. And $y_{ij}$ denotes the corresponding

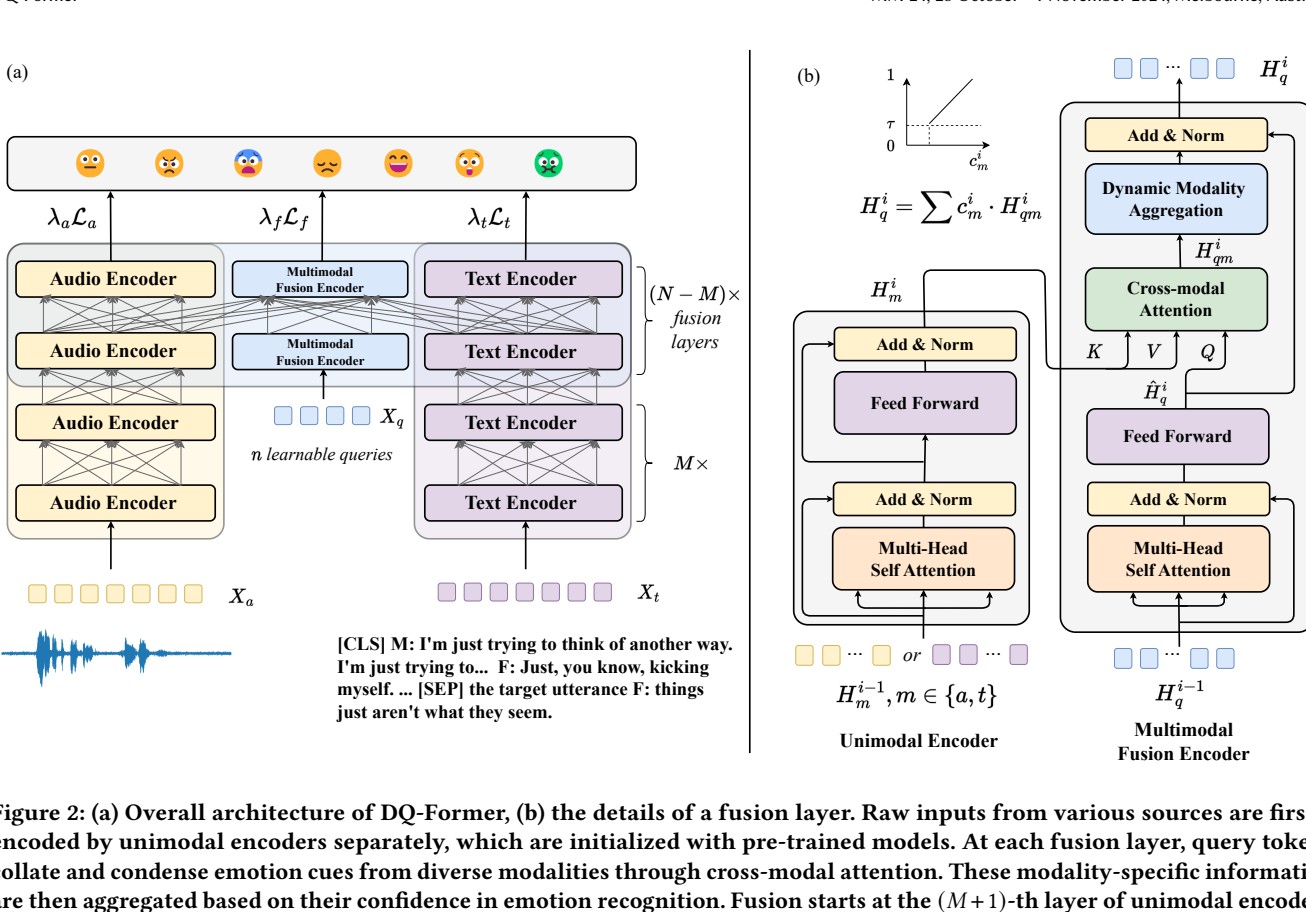

Figure 2: (a) Overall architecture of DQ-Former, (b) the details of a fusion layer. Raw inputs from various sources are firstly encoded by unimodal encoders separately, which are initialized with pre-trained models. At each fusion layer, query tokens collate and condense emotion cues from diverse modalities through cross-modal attention. These modality-specific information are then aggregated based on their confidence in emotion recognition. Fusion starts at the $(M+1)$-th layer of unimodal encoders, with the initial $M$ layers serving as unimodal feature extractors.

emotion label of utterance $u_{ij}$, which belongs to $k$ emotion categories. This task aims to identify the emotion label $y_{ij}$ of each utterance $u_{ij}$ from multiple types of signals.

### 3.2 Unimodal Encoder

Regarding the multimodal input, each utterance consists of text transcripts ($t$) and audio ($a$). As highlighted by Mao et al. [24], textual modality performs much better in context-dependent settings, while acoustic modality always achieve better performance in context-free settings. Consequently, we structure our raw transcripts by incorporating dialogue history, as in "*<cls> n preceding (speaker: utterance) pairs <sep> the target (speaker: utterance)*", while maintaining audio inputs as context-free.

Formally, we adopt $m$ to represent a particular input modality, where $m \in \{t, a\}$ can be either $t$ for textual modality or $a$ for acoustic modality. Let's denote the input sequence as $X_m = \{x_1, x_2, \ldots, x_{N_m}\}$, where $N_m$ signifies the length of the input sequence for modality $m$.

To obtain modality-specific attributes and model intra-modal interactions, we encode the raw inputs from different modalities with separate unimodal transformer encoders. These unimodal encoders are initialized with pre-trained models, allowing them to inherit knowledge from pre-training stages. The textual encoder is initialized by BERT model [4], while acoustic encoder is initialized using

WavLM model [3]. The unimodal intermediate representations can be expressed as follows:

$$H_m^i = \mathcal{E}_m^i(H_m^{i-1}), \text{ and } H_m^0 = \mathcal{M}_m(X_m) \quad (1)$$

Here, $\mathcal{E}$ is the vanilla Transformer encoder layer, comprising a multi-head self-attention mechanism and a feed-forward network [32], depicted as the left portion of Figure 2 (b). The operator $\mathcal{M}$ signifies a linear projection that maps each token to $\mathbb{R}^d$. $H_m^0$ denotes the initial token embedding representations, while $H_m^i$ represents the hidden representation at the $i$-th encoder layer for modality $m$.

### 3.3 Multimodal Fusion Encoder

Cognitive-related studies indicate that emotional expressions are better identified through specific attributes rather than broad semantic aspects [13]. Meanwhile, recent research on modality-specific models suggests that rich hierarchical information exists across different layers, with semantic features in higher layers and more modality-specific attributes in intermediate layers [3, 5]. Hence, we opt for a middle-fusion approach instead of traditional late-fusion methods to capture diverse levels of emotional information.

Inspired by Nagrani et al. [25] and Li et al. [15], we adopt a set of learnable queries to gather and condense emotion-relevant attributes from each modality within intermediate layers. Figure

2 (b) provides a detailed illustration of the multimodal fusion encoder layer. Specifically, we randomly initialize the query tokens $X_q = \{q_1, q_2, \ldots, q_l\}$. And we intentionally set the length $l$ to be much shorter than that of unimodal inputs, which aims to condense information and mitigate computational complexity. These query tokens serve as the input of the multimodal fusion encoder. They firstly interact with each other through self-attention like unimodal encoders do:

$$\hat{H}_q^i = \mathcal{E}_q^i(H_q^{i-1}), \text{ and } H_q^0 = \mathcal{M}_q(X_q) \tag{2}$$

Subsequently, we employ cross-modal attention [31] to project information from various modalities into a unified query space, resembling a shared multimodal space. This approach aims to maximize synergy and minimize heterogeneity gaps among diverse modalities. Assume that the query representations $\hat{H}_q^i$ is the $Q$, and the unimodal hidden representations $H_m^i$ is the $K$ and $V$. The equation defining this process is:

$$H_{qm}^i = C_{m \to q}^i(\hat{H}_q^i, H_m^i, H_m^i) \tag{3}$$

Here, $C_{m \to q}^i$ represents the cross-modal attention. The detailed calculation of $C_{m \to q}$ is as follows:

$$C_{m \to q} = \mathcal{LN}(Q + \mathcal{F}(\mathcal{LN}(Q + \text{softmax}(\frac{QK^T}{\sqrt{d}})V))) \tag{4}$$

where $\mathcal{LN}$ denotes **LayerNorm**, and $\mathcal{F}$ is a **FFN** network.

Finally, we aggregate $H_{qt}^i$ and $H_{qa}^i$ to obtain $H_q^i$ through dynamic modality priority learning, which will be detailed in Session 3.4. After multiple layers of inter-modal interaction and fusion, the final representation of query tokens can be regarded as the multimodal representation for emotion recognition.

## 3.4 Dynamic Modality Priority Learning

Intuitively, not all modalities play an equal role in emotion recognition. The goal of the instance-level dynamic modality priority learning is to mask unnecessary modalities before fusion and dynamically control the fusion degree of each modality.

To enable dynamic modality fusion, we add a confidence estimation network (ConNet). The ConNet takes hidden states of the unimodal encoder at the $i$-th layer as inputs and predicts a single scalar between 0 and 1. The confidence calculation is as follows.

$$h_m^i = \text{AvgPool}(H_m^i) \tag{5}$$

$$\tilde{c}_m^i = \sigma(Wh_m^i + b) \tag{6}$$

$$c_m^i = \frac{\tilde{c}_m^i}{\sum_{j \in \{a,t\}} \tilde{c}_j^i} \tag{7}$$

where $W$ and $b$ are trainable parameters, $\sigma(\cdot)$ is the sigmoid function.

For each instance, the ConNet outputs a high confidence score if the attribute of modality $m$ at $i$-th layer is a crucial emotion cue. Conversely, if this attribute is unimportant, it outputs a low confidence score. When the confidence score $c_m^i$ is close to 0, we can assume that the modality is unnecessary or even a noisy signal that may mislead the final emotion recognition result. Therefore, we set a mask threshold $\tau$ (i.e., $\tau = 0.1$) to filter out unnecessary modalities. After applying the masking operation of misleading

**Table 1: Statistics of IEMOCAP and MELD datasets.**

| Statistics | IEMOCAP | | MELD | |
|---|---|---|---|---|
| | # dialogues | # utterances | # dialogues | # utterances |
| Train | 121 | 5,929 | 1,039 | 9,989 |
| Dev | 14 | 600 | 114 | 1,109 |
| Test | 16 | 851 | 280 | 2,610 |
| Total | 151 | 7,380 | 1,433 | 13,708 |

modalities, $c_m^i$ is updated as follows:

$$c_m^i = \begin{cases} c_m^i, & if \ c_m^i \geq \tau \\ 0, & otherwise \end{cases} \tag{8}$$

This ConNet is inserted in every fusion layer. Finally, the aggregation function of $H_q^i$ is as follows:

$$H_q^i = \sum c_m^i \cdot H_{qm}^i \tag{9}$$

## 3.5 Optimization Object

Our DQ-Former framework comprises two key components: unimodal encoders facilitating intra-modal interaction, and a multimodal fusion encoder enabling inter-modal interaction. We denote the representations of the final layer as unimodal representations and multimodal representations:

$$h_a = \text{AvgPool}(H_a^N) \tag{10}$$

$$h_t = \text{AvgPool}(H_t^N) \tag{11}$$

$$h_f = \text{AvgPool}(H_q^N) \tag{12}$$

Here, $a$, $t$, and $f$ correspond to *fusion*, *textual* and *acoustic*, respectively. These representations are utilized to obtain the final unimodal and fusion predictions:

$$y_\alpha = W_\alpha h_\alpha + b_\alpha \tag{13}$$

where $W_\alpha \in \mathbb{R}^{k \times d}$, $b_\alpha \in \mathbb{R}^k$, $\alpha \in \{a, t, f\}$ and $k$ represents the number of emotion labels.

Our training object involves two parts: cross-entropy loss for unimodal predictions and multimodal predictions. The training optimization objective is expressed as:

$$\mathcal{L} = \lambda_f \mathcal{L}_f + \lambda_t \mathcal{L}_t + \lambda_a \mathcal{L}_a, \tag{14}$$

where $\lambda_f$, $\lambda_t$ and $\lambda_a$ are the weighting factors. During inference, the fusion prediction $y_f$ is chosen.

## 4 EXPERIMENTAL SETTING

### 4.1 Experimental Dataset

To evaluate the efficacy of our model, we conduct experiments on two widely recognized MERC benchmark datasets: IEMOCAP [2] and MELD [27]. The IEMOCAP dataset is a multimodal emotion recognition dataset, which contains 151 conversations and 10,039 utterances. Each utterance is annotated with predefined emotion labels such as anger, happiness, sadness, neutral, excitement, frustration, fear, surprise, and others. Due to the data imbalance issue, we only focus on the first six main emotion categories, excluding the remaining labels. The MELD dataset is a multi-party conversational dataset collected from *the Friends* TV shows. This dataset contains

13,708 utterances and 1,433 conversations. Each utterance is annotated with one of the following seven labels: neutral, surprise, fear, sad, joy, disgust, and angry.

Since the IEMOCAP dataset does not provide an official data split, we split all the conversations into training, testing, and validation sets according to a ratio of 8:1:1, ensuring that each dialogue appears in only one of the data divisions to avoid contextual content leakage. For MELD datasets, we follow the default data splitting. A concise summary of statistical values for all datasets is provided in Table 1.

## 4.2 Implementation Details

We train two variants of DQ-Former, including base and large version. The DQ-Former base comprises 12 unimodal encoder layers, 4 multimodal fusion layers (fusion starts at the 9-th layer of the unimodal encoder), 8 attention heads, and hidden dimensions of 768. In the case of DQ-Former large, there are 24 unimodal encoder layers, and a hidden dimension of 1024. The textual and acoustic modality encoders are initialized using BERT [4] and WavLM [3] pretrained models, respectively. As for input of the multimodal fusion transformer, we set 16 learnable query tokens, which are randomly initialized. For the textual modality, we adapt a dialogue history window size of 5 utterances, restricting the input length to 512 tokens. For the acoustic modality, we set a limit of 10 seconds for the duration of audio input. Due to the imbalance across various emotions, we also perform up-sampling on minority class data while training. Besides, we set the loss weights $\lambda_f$, $\lambda_t$, and $\lambda_a$ to 1.

All experiments are conducted using a single Tesla V100 GPU. To optimize trainable parameters, we use the AdamW optimization scheme [21]. The learning rate is set to 1e-5, and it follows a linear warm-up of 1% of total steps. We set a batch size of 8. The training process spans 20 epochs. To mitigate overfitting, we incorporate an early stopping mechanism with a patience of 7 evaluation steps. Subsequently, we employ the checkpoints that performed best on the validation set for testing. We run each experiment three times and report the average result.

Consistent with previous studies, we utilize the Weighted Average F1 score (WAF1) as the main metric, considering the natural imbalance across various emotions. Weighted Average Accuracy (WAA) serves as the secondary evaluation metric.

## 4.3 Baseline Models

To validate the performance of DQ-Former, we conduct a comprehensive comparison with several state-of-the-art models. The comparative models are as follows: **MMGCN** [11], **DialogTRM** [24], **MM-DFN** [10], **EmoCaps** [19], **FacialMMT**[41], **GA2MIF** [16], **Joyful** [14]. All these models leverage textual, acoustic, and visual modalities as inputs. Additionally, we include **CMCF-SRNet** [40] in our baselines, as it is the advanced two-modality model incorporating textual and acoustic inputs.

Since the choice of unimodal features seriously affects multimodal results, we also reproduce the following models under the same experimental settings as ours: **MulT** [31] (a conventional late-fusion framework for unaligned inputs), **MBT** [25] (a novel middle-fusion framework), and **SA** (it combines different modalities into unified inputs for the single-stream transformer). The unimodal encoders in these three models are also tuned.

## 5 RESULTS AND DISCUSSION

To validate the efficacy of our approach, we first conduct comparative experiments with currently advanced systems. Following this, we perform ablation experiments. Finally, we discuss the interpretability of our model, examining how DQ-Former can extract necessary emotional cues across various layers within each modality and explicitly discern the relative importance of different modalities.

## 5.1 Comparison Results with Existing Works

We compare our proposed method with state-of-the-art approaches on the MELD and IEMOCAP datasets, and present the results in Table 2. The results show that DQ-Former demonstrates competitive performance on both two datasets, particularly in the scenario involving only textual and acoustic modalities. Specifically, on the IEMOCAP dataset, DQ-Former (base) demonstrates the most promising performance, achieving a significant relative improvement of 1.43% WF1 compared to the second-best result obtained by EmoCaps. Similarly, on the MELD dataset, DQ-Former (large) outperforms FacialMMT, while DQ-Former (base) also achieves competitive results.

Additionally, DQ-Former showcases outstanding performance in recognizing certain minority emotion categories. For example, both DQ-Former (base) and DQ-Former (large) demonstrate excellent emotion recognition capabilities in the fear and disgust categories on the MELD dataset. This highlights the effectiveness of masking misleading modalities to achieve more robust performance.

It's worth noting that, under identical experimental conditions, the new middle-fusion framework DQ-Former consistently outperforms MulT and MBT. Compared to the typical late-fusion MERC system MulT, DQ-Former (base) achieves substantial improvements by 2.91% and 6.51% WF1 on the MELD and IEMOCAP datasets, respectively. Furthermore, integrating DMP results in significant performance enhancement compared to MBT, another middle-fusion method. This underscores the critical role of leveraging a dynamic modality priority fusion strategy to integrate diverse granularities of emotion cues from different modalities across various intermediate layers.

Moreover, it's observed that DQ-Former (large) does not always outperform DQ-Former (base). This phenomenon may be attributed to the limited existing data combined with the high parameter count of the model, potentially leading to overfitting issues.

## 5.2 Ablation Studies

We conduct experiments to shed light on the following reasonable doubts:

- Does middle-fusion strategy outperform other fusion strategies?
- Can dynamic modality priority learning (DMP) decide the relative importance between textual and acoustic modalities and can mask threshold effectively filter noising modalities?
- How does DQ-Former perform under modality missing circumstance?

In addition to validating on the IEMOCAP 6-way condition, we introduce the IEMOCAP 4-way condition, which concentrates

**Table 2: Overall comparison results on IEMOCAP and MELD datasets. The best result is highlighted in bold. ♮ indicates our reproduce results under the same setting as ours, while other results are from the original paper. "*" indicates scenarios with only textual and acoustic modalities, while others takes account of three modalities.**

| Method | MELD | | | | | | | | | IEMOCAP 6-way | | | | | | | |
| | Neu. | Joy. | Sad. | Ang. | Fear | Dis. | Sur. | WAA | WAF1 | Neu. | Hap. | Sad | Ang. | Exc | Fru. | WAA | WAF1 |
|---|---|---|---|---|---|---|---|---|---|---|---|---|---|---|---|---|---|
| MMGCN | 76.33 | 53.02 | 26.74 | 46.09 | — | — | 48.15 | 59.31 | 58.31 | 64.36 | 45.14 | 77.16 | 68.82 | 74.71 | 61.40 | 66.36 | 66.26 |
| MM-DFN | 77.76 | 54.78 | 22.94 | 47.82 | — | — | 50.69 | 62.49 | 59.46 | 66.42 | 44.22 | 78.98 | 69.77 | 75.56 | 66.33 | 68.21 | 68.18 |
| DialogueTRM | — | — | — | — | — | — | — | **65.70** | 63.50 | — | — | — | — | — | — | 69.50 | 69.70 |
| EmoCaps | 77.12 | 57.50 | **42.50** | **57.54** | 3.03 | 7.69 | 63.19 | — | 64.00 | 64.48 | **71.91** | **85.06** | 68.99 | 78.41 | 66.76 | — | 71.11 |
| GA2MIF | 76.92 | 51.87 | 27.18 | 48.52 | — | — | 49.08 | 61.65 | 58.94 | 68.38 | 46.15 | 84.50 | 70.29 | 75.99 | 66.49 | 69.75 | 70.00 |
| Joyful | 76.80 | 56.89 | 41.78 | 50.71 | — | — | 63.53 | 61.77 | 68.24 | 60.94 | 84.42 | 69.95 | 73.54 | 67.55 | 70.55 | 71.03 | |
| FacialMMT | 78.55 | **61.10** | 38.51 | 53.66 | 13.04 | **30.30** | 58.17 | — | 64.69 | — | — | — | — | — | — | — | — |
| CMCF-SRNet* | — | — | — | — | — | — | — | — | 62.30 | 68.80 | 52.20 | 80.90 | 70.30 | 76.70 | 61.60 | 70.50 | 69.60 |
| SA*♮ | 72.61 | 48.99 | 17.73 | 46.61 | — | — | 42.14 | 57.65 | 54.60 | 46.84 | 43.51 | 60.40 | 73.61 | 62.85 | 56.49 | 57.86 | 57.24 |
| MulT*♮ | 78.76 | 57.41 | 34.52 | 47.67 | — | — | 55.55 | 63.94 | 61.77 | 60.41 | 59.82 | 66.30 | 77.50 | 76.29 | 60.53 | 66.40 | 66.03 |
| MBT*♮ | 78.39 | 56.44 | 34.24 | 48.56 | 3.85 | 9.41 | 56.61 | 63.76 | 61.98 | 54.51 | 57.39 | 68.78 | 74.33 | 76.72 | 59.57 | 64.84 | 64.26 |
| DQ-Former(base)* | **78.86** | 59.86 | 35.71 | 51.16 | 19.38 | 22.36 | 57.89 | 64.95 | 63.96 | 69.18 | 64.09 | 69.92 | **79.74** | **83.39** | **69.48** | **72.63** | **72.54** |
| DQ-Former(large)* | 78.31 | 60.84 | 39.93 | 52.82 | **21.90** | 26.54 | 59.13 | 64.88 | **64.70** | **71.87** | 61.67 | 69.61 | 77.39 | 80.95 | 67.71 | 71.68 | 71.76 |

**Table 3: Comprehensive ablation results on IEMOCAP and MELD.** $\tau$ **is the mask threshold.** *w/o* $\tau$ **indicates the absence of this threshold, while** *w/o DMP* **signifies the exclusion of dynamic modality priority learning.** *late* **and** *early* **denote fusion starting from the last layer and from the first layer, respectively.**

| Models | MELD | | IEMOCAP 6-way | | IEMOCAP 4-way | |
| | WAA | WAF1 | WAA | WAF1 | WAA | WAF1 |
|---|---|---|---|---|---|---|
| $\tau = 0.1$ | **64.95** | **63.96** | **72.63** | **72.54** | **87.58** | **87.59** |
| $\tau = 0.2$ | 64.60 | 62.64 | 72.09 | 71.81 | 85.95 | 85.87 |
| w/o $\tau$ | 64.17 | 62.75 | 71.95 | 72.00 | 86.76 | 86.72 |
| w/o DMP | 63.91 | 62.26 | 69.88 | 69.47 | 86.15 | 86.22 |
| late | 64.08 | 62.57 | 70.19 | 70.09 | 83.50 | 83.07 |
| early | 63.57 | 63.09 | 71.73 | 70.90 | 87.17 | 87.17 |

solely on the four primary emotions: anger, happiness, sadness, and neutral. Ablation results are presented in Table 3.

*5.2.1 Role of Middle-fusion Strategy.* DQ-Former employs a middle-fusion strategy to effectively leverage various levels of modality-specific emotional features embedded in intermediate layers of pre-trained unimodal encoders. To delve deeper into the impact of these fusion strategies, we introduce two variants of DQ-Former: DQ-Former (late) which uses the final representation to fuse, and DQ-Former (early) which starts fusion at the first layer.

Results in Table 3 consistently demonstrate the superiority of middle-fusion over both late-fusion and early-fusion approaches. Notably, the late-fusion strategy exhibits a clear performance decline. Specifically, DQ-Former (late) shows a decrease of 1.39%, 2.45% and 4.52% WF1 on MELD, IEMOCAP 6-way and IEMOCAP 4-way, respectively. In contrast, DQ-Former (early) exhibits relatively less performance degradation. This emphasizes the importance of adaptively considering the various levels of emotional information across different layers within each modality. Moreover, it aligns with the findings of cognitive-related research, indicating that human emotions are more easily recognized through modality-specific attributes.

*5.2.2 Role of Dynamic Modality Priority Learning.* To further investigate the impact of the DMP module in DQ-Former, we conduct a comprehensive ablation analysis. Specifically, we vary the mask threshold to 0.2 ($\tau$=0.2) and 0 (w/o $\tau$), and also remove the DMP module entirely (w/o DMP). From the results, we observe:

*Firstly*, the removal of DMP resultes in a significant performance decrease, with a drop of 1.7%, 3.07% and 1.37% WF1 on MELD, IEMO-CAP 6-way and IEMOCAP 4-way, respectively. This underscores the critical role of DMP in enhancing effective multimodal fusion. *Secondly*, masking misleading modalities effectively enhances the robustness of multimodal emotion recognition. DQ-Former (w/o $\tau$) experiences a performance decrease of 0.54% WF1 on IEMOCAP 6-way, 0.87% WF1 on IEMOCAP 4-way and 1.21% WF1 on MELD. *Moreover*, it is essential to carefully select the mask threshold. Setting $\tau$ to 0.2 leads to a performance decline on both two datasets. This decline may be attributed to the high mask threshold, which could potentially filter out useful signals, treating them as noise.

*5.2.3 Role of Each Modality and Robustness Analysis.* To further investigate the role of each modality in MERC, we fine-tune the unimodal encoders on respective modality data, referred to as WavLM-FT and BERT-FT. Additionally, we assess the robustness of DQ-Former (base) under modality-missing scenarios. Specifically, we simulate the absence of a modality by masking it (i.e., setting the missing modality inputs to 0). From the results in Table 4 we find:

Firstly, the multimodal fusion results of DQ-Former consistently outperform the unimodal setups across both the MELD and IEMO-CAP datasets, demonstrating its capability in leveraging complementary information from diverse modalities.

Moreover, under modality-missing circumstances, all three models experience notable performance degradation, particularly evident when the textual modality is missing. For example, SA suffers a performance drop of over 50% when text inputs are masked, indicating a phenomenon of modality collapse during training. In contrast, DQ-Former demonstrates superior robustness to modality-missing, which maintains comparable performance to fine-tuned unimodal encoders. It is worth noting that if a multimodal system overly depends on the dominant modality, it will become less robust in scenarios where one modality is missing.

In addition, not all datasets are suitable for MERC. On the MELD dataset, BERT-FT achieves 60.94% WF1, while WavLM-FT reaches 31.26%. Interestingly, all three models exhibit a slight performance

**Table 4: Robust ablation results on MELD and IEMOCAP datasets. ‡ indicates masking audio inputs, while † denotes masking text inputs. The red results indicate the percentage decrease in model performance relative to the original model when one modality is missing, while the green ones indicate an increase.**

| Method | MELD | | IEMOCAP 6-way | |
|---|---|---|---|---|
| | WAA | WF1 | WAA | WF1 |
| BERT-FT | 61.65 | 60.94 | 67.48 | 66.98 |
| WavLM-FT | 47.82 | 31.26 | 52.85 | 51.92 |
| MulT | 63.94 | 61.77 | 66.40 | 66.03 |
| MulT‡ | 63.92 | 62.31 | 61.72 | 60.52 |
| | -0.04 | +0.87 | -7.04 | -8.34 |
| MulT† | 48.12 | 31.27 | 36.96 | 30.54 |
| | -24.75 | -49.38 | -44.34 | -53.74 |
| SA | 57.65 | 54.60 | 57.86 | 57.24 |
| SA‡ | 57.87 | 54.23 | 47.38 | 45.58 |
| | +0.38 | -0.68 | -18.10 | -20.38 |
| SA† | 22.24 | 19.29 | 26.78 | 21.47 |
| | -61.43 | -64.68 | -53.71 | -62.50 |
| DQ-Former | 64.95 | 63.96 | 72.63 | 72.54 |
| DQ-Former‡ | 65.58 | 64.06 | 68.50 | 68.23 |
| | +0.96 | +0.16 | -6.04 | -6.41 |
| DQ-Former† | 49.16 | 39.05 | 51.29 | 50.81 |
| | -24.32 | -38.96 | -29.65 | -30.31 |

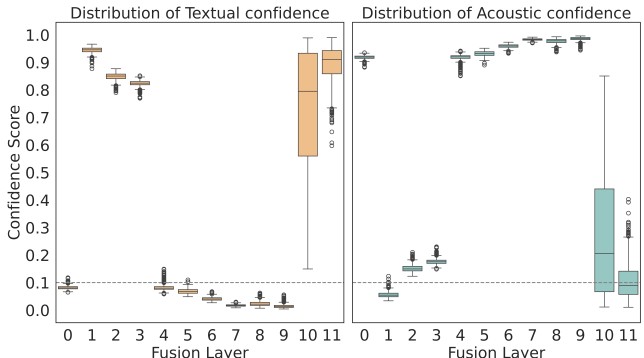

**Figure 3: Modality confidence distribution across different layers on the IEMOCAP dataset. The mask threshold is set to 0.1, as indicated by the dashed line. Middle layers serve as feature selectors, while top layers predominantly encapsulate semantic and content-level information and there is a priority trade-off between these two modalities.**

improvement trend when audio inputs are masked. This means the text inputs dominate in the MELD dataset. Since the MELD dataset is collected from *the Friends* TV show, the presence of background noise in the audio may hinder emotion recognition performance.

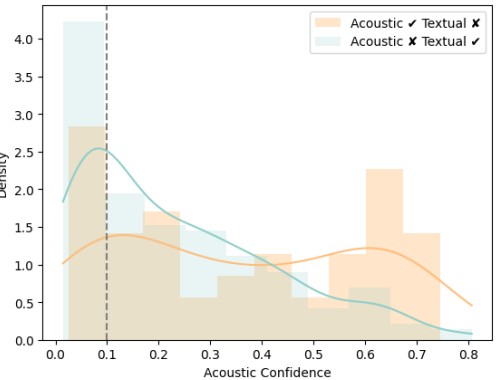

**Figure 4: Acoustic confidence distribution in the penultimate layer for two categories of samples in the IEMOCAP dataset: *Text-Dominated* and *Audio-Dominated*. The x-axis shows acoustic confidence scores, and the y-axis indicates the amount of samples for each category. Density estimation is conducted using Gaussian Kernel Density Estimation. The mask threshold is depicted by the dashed line.**

### 5.3 Interpretability Discussions

*5.3.1 The Modalities Priority across Different Intermediate Layers.* DQ-Former fuses at intermediate layers through DMP to integrate various levels of emotional features of the unimodal pre-trained model. To further investigate the priority of modalities across the layers, we examine the distribution of confidence scores across fusion layers of the DQ-Former (early). The result is depicted in Figure 3.

The result reveals several discernible patterns: **1)** The textual modality predominantly asserts its influence in the lower layers, with confidence scores peaking between 0.8 and 0.9. **2)** In contrast, the acoustic modality takes precedence in the middle layers, with confidence scores close to 1. **3)** As we ascend to the higher layers, the confidence score distribution for both modalities becomes more widespread, indicating a delicate balance with a slight inclination towards the textual modality. This underscores that our model adaptively selects key acoustic attributes and textual semantic features for emotion recognition, which is aligned with human perception processes.

*5.3.2 The Modalities Priority of Different Instances.* As introduced in section 3.4, the goal of DMP is to selectively mask misleading modalities before fusion, while dynamically managing modality fusion priorities for every certain instance. To further study the effectiveness of DMP, we examine the distribution of acoustic confidence scores across diverse instances.

We select two distinct categories of samples from IEMOCAP, focusing on instances correctly classified by DQ-Former: 1) *Text-Dominated*: These samples are correctly classified by the textual modality but are misclassified by the acoustic modality. 2) *Audio-Dominated*: These samples are correctly classified by the acoustic modality but are misclassified by the textual modality. We analyze the confidence distribution in the penultimate layer of the multimodal encoder, considering that top layers primarily contain

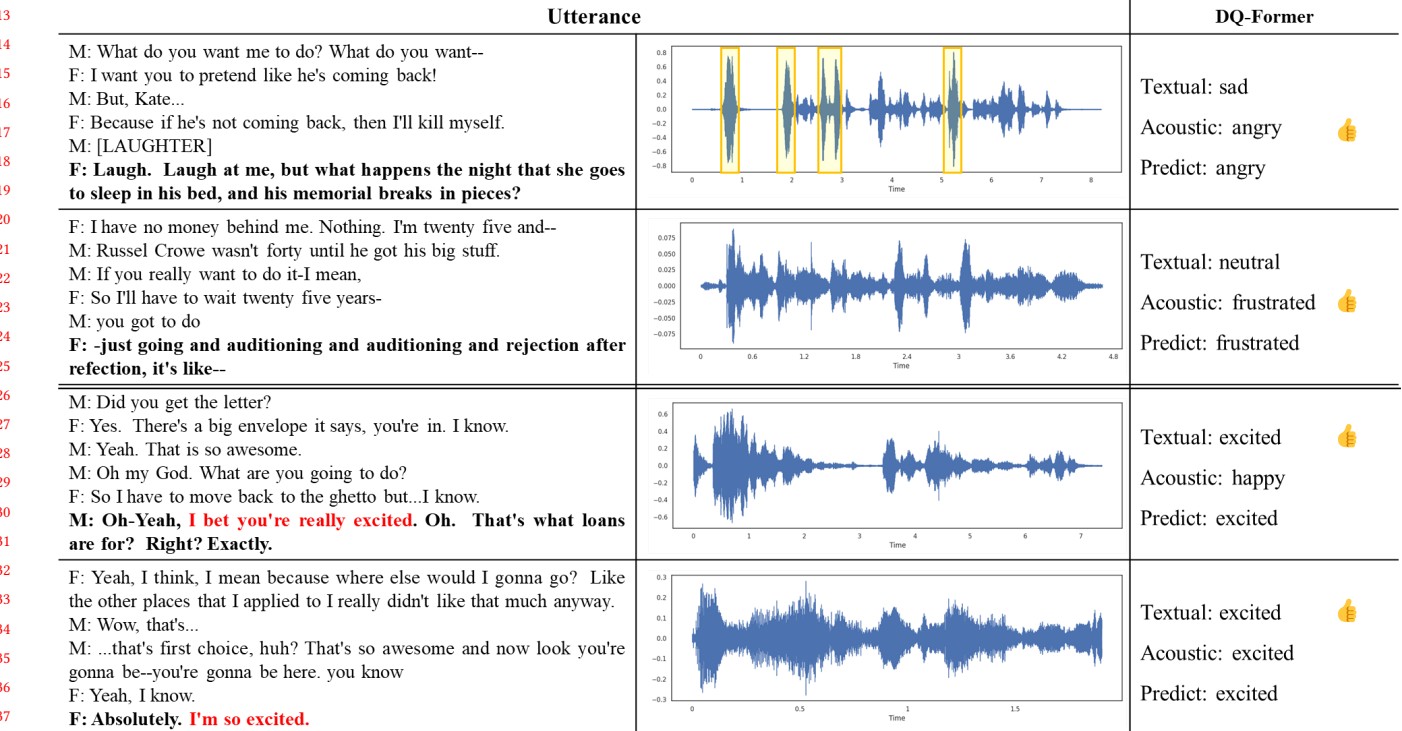

**Figure 5: Case study illustrating the performance of DQ-Former. Four cases are sampled from the IEMOCAP dataset, with the first two representing Audio-Dominated samples and the last two representing Text-Dominated samples.**

semantic and content-level information, with a trade-off between different modalities.

Figure 4 presents the acoustic confidence distribution for both categories. Remarkably, for Audio-Dominated samples, confidence mainly exceeds 0.1, whereas for Text-Dominated ones, it peaks between 0 and 0.1, gradually decreasing with higher confidence. This trend mirrors human cognitive tendencies, indicating higher acoustic confidence in Audio-Dominated instances and lower acoustic confidence in Text-Dominated ones. Refer to Session 5.4 for additional practical examples.

Additionally, we observe that when the acoustic confidence score is below the mask threshold ($\tau = 0.1$), the number of Text-Dominated samples significantly exceeds other confidence intervals, while the count of Audio-Dominated samples is notably lower in this interval compared to others. This observation underscores the necessity of filtering out misleading modalities.

### 5.4 Case Study

To better understand the main contributions of our work, we present four examples in Figure 5. These examples are drawn from Audio-Dominated and Text-Dominated categories in the IEMOCAP dataset. The results delineate distinctive characteristics between Audio-Dominated and Text-Dominated samples. In the former, overt vocal cues are prominently apparent. For instance, in case 1, a waveform graph is characterized by heightened amplitude, indicative of palpable anger expressed by the speaker. Conversely, case 2 showcases

subdued amplitude and specific inflection, reflective of frustrated emotions. In contrast, Text-Dominated samples usually contain emotional vocabulary that clearly conveys the speaker's emotions. These observations underscore the efficacy of the DMP module in DQ-Former, which adaptively captures and integrates necessary emotional cues from diverse modalities.

## 6 CONCLUSION

In this paper, we introduce DQ-Former, a novel middle-fusion multimodal framework designed to enhance the modeling of diverse emotion cues for a robust MERC system. DQ-Former leverages various granularities of emotional information embedded in intermediate layers within each modality, and gathers them by learning the confidence levels associated with emotion recognition for complementary insights. Experiments on the IEMOCAP and MELD datasets demonstrate the effectiveness of DQ-Former in improving emotion recognition performance. Furthermore, our in-depth analysis demonstrates the ability of DQ-Former to extract key emotional cues within each modality and autonomously adjust the priority of each modality under different scenarios. This closely aligns with human cognitive processes and achieves robust and interpretable emotion recognition results. These findings offer valuable insights for leveraging advanced modality-specific pre-trained models from a cognitive perspective, thereby informing and potentially improving future research endeavors in multimodal emotion recognition.

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
