# OpenReview forum: "DQ-Former: Querying Transformer with Dynamic Modality Priority for Cognitive-aligned Multimodal Emotion Recognition in Conversation"
_acmmm.org/ACMMM/2024/Conference — MM2024 Poster_

### Official Review · Reviewer_hvvB · 2024-05-20

**Rating:** 3
**Confidence:** 3

**Summary:**

- The author proposes a cognitive-aligned middle-fusion multimodal framework called DQ-Former to enhance the modeling of emotional cues for a robust MERC system.
- This framework includes a dynamic modality priority learning module to aggregate text and audio inputs, managing the contribution of each modality for every instance. Experiments were conducted on the MELD and IEMOCAP datasets.

**Strengths:**

1. **Methods**:

- The author opts for a middle-fusion approach instead of traditional late-fusion methods, capturing diverse levels of emotional information by initializing query tokens with a shorter length than unimodal inputs. These tokens serve as inputs to the multimodal fusion encoder, allowing them to interact through self-attention. Cross-modal attention is employed to project information from various modalities.
- Dynamic Modality Priority Learning aims to mask unnecessary modalities before fusion and dynamically control the fusion degree of each modality.
2. **Experiments**:
- The experiments are extensive, with comparisons to various previous works.
- The ablation study sufficiently demonstrates the effectiveness of DQ-Former and its components.
3. **Structure and Writing**:
The paper is well-written and easy to follow. The literature review is systematic and comprehensive.

**Limitations:**

**Limitations:**

1. **Methods:**
- The concept of middle-fusion needs clarification, especially in comparison to deep-fusion or layer-wise fusion in multimodal learning. Additional related works should be included to highlight the novelty of the proposed method.

2. **Experiments:**
- DQ-Former only explores text and audio modalities, comparing them to baseline models that use all three modalities, which is not entirely fair. Although audio+text provides strong results in MER, aligning all three modalities is challenging. The author should compare with SOTA baseline models that use only text and audio for fairness, citing and adding results from relevant papers (e.g., papers [1-4]).
- Additionally, for methods using all three modalities, comparison with more advanced SOTA models is needed (e.g., studies [5-8]).
- Table 2 compares the proposed model with baseline models. The author states that only SA, MulT, and MBT are reproduced, while others are from original papers. **However, the reported results do not match those in the original papers. This needs serious review as it affects the paper's reliability**.
- The experimental section lacks significance tests to verify the method's effectiveness further.

3. **Reproducibility:**
- The provided source code facilitates reproducibility.

4. **Typos, Grammar, Style, and Presentation:**
- Some quantities need clearer dimensionality specification for better model flow understanding.

5. **Questions:**
- Does the random creation of query tokens with varying lengths and continuous fusion increase computational complexity? Has the author measured this compared to simpler fusion methods? If so, measurable quantities like complexity, runtime per epoch, and parameter count should be provided.
- In Table 4, DQ-Former and SA results for the MELD dataset improve when audio input is masked. Can the author explain this phenomenon?

**References**

[1] Yoon, Seunghyun, et al. "Speech emotion recognition using multi-hop attention mechanism." ICASSP 2019-2019 IEEE International conference on acoustics, speech and signal processing (ICASSP). IEEE, 2019.

[2] Peng, Zixuan, et al. "Efficient speech emotion recognition using multi-scale cnn and attention." ICASSP 2021-2021 IEEE international conference on acoustics, speech and signal processing (ICASSP). IEEE, 2021.

[3] Feng, Lin, et al. "Multimodal speech emotion recognition based on multi-scale MFCCs and multi-view attention mechanism." Multimedia Tools and Applications 82.19 (2023): 28917-28935.

[4] Khan, Mustaqeem, et al. "MSER: Multimodal speech emotion recognition using cross-attention with deep fusion." Expert Systems with Applications 245 (2024): 122946.

[5] Hu, Guimin, et al. "UniMSE: Towards Unified Multimodal Sentiment Analysis and Emotion Recognition." Proceedings of the 2022 Conference on Empirical Methods in Natural Language Processing. 2022.

[6] Nguyen, Cam Van Thi, et al. "Conversation Understanding using Relational Temporal Graph Neural Networks with Auxiliary Cross-Modality Interaction." Proceedings of the 2023 Conference on Empirical Methods in Natural Language Processing. 2023.

[7] Shi, Tao, and Shao-Lun Huang. "MultiEMO: An attention-based correlation-aware multimodal fusion framework for emotion recognition in conversations." Proceedings of the 61st Annual Meeting of the Association for Computational Linguistics (Volume 1: Long Papers). 2023.

[8] Tu, Geng, et al. "Adaptive Graph Learning for Multimodal Conversational Emotion Detection." Proceedings of the AAAI Conference on Artificial Intelligence. Vol. 38. No. 17. 2024.

**Suitability:**

3

---

### Official Review · Reviewer_oqoh · 2024-05-25

**Rating:** 2
**Confidence:** 2

**Summary:**

This paper proposes a new multimodal fusion framework for emotional recognition. Unlike prior works, it incorporates various levels of emotional features within each modality and considers the unique contributions of different modalities. Concretely, it suggests a new middle-fusion encoder with cross-modal attention and dynamic modality priority learning with threshold. The experimental results demonstrate the effectiveness of proposed methods on IEMOCAP and MELD datasets.

**Strengths:**

[S1. New methodology for middle-fusion] This paper suggests a new middle-fusion framework to both capture the diverse degree of emotional features and reflect the different contribution of each modality. Specifically, it opts for middle-fusion and adopts cross-attention with learnable query. It also dynamically aggregates the modalities at the instance level by utilizing ConNet.

[S2. Effectiveness] The proposed method improves the emotion recognition performance in two conversational datasets. It also demonstrates robustness in modality-missing scenarios.

**Limitations:**

[W1. Unclear comparison to baselines] This paper does not clarify its contribution compared to baselines of the same intuitions. Specifically, this paper suggests two intuitions 1) incorporating the various levels of emotional features within each modality and 2) considering the different priorities of each modality. However, these intuitions are already introduced in baselines as described in Section 2. Thus, it is necessary to compare this paper and baselines in terms of each intuition, and show why the proposed framework has shown the enhancement.


[W2. Insufficiently supported analysis] Experimental results are not supportive of the analysis. For example, in section 2.3.2, the distribution of audio-dominated samples seems not appropriate. This is because the DMP should have assigned higher acoustic confidence in the most of the audio-dominated samples. Also, the insufficient experiment in Table 2 does not support the statement that DQ-former shows recognition capability in minor categories (e.g. fear, disgust).


[W3. Weak baselines] Baselines are too weak. For example, in robustness analysis(5.2.3), the baselines are weak because most baselines are published before 2022. In the case of MELD datasets results, there are some models that show better performance, UniMSE[1], DF-ERC[2]. It is necessary to compare these models.



[1] Song, X., Huang, L., Xue, H., & Hu, S. (2022). Supervised prototypical contrastive learning for emotion recognition in conversation. arXiv preprint arXiv:2210.08713.
[2] Li, B., Fei, H., Liao, L., Zhao, Y., Teng, C., Chua, T. S., ... & Li, F. (2023, October). Revisiting disentanglement and fusion on modality and context in conversational multimodal emotion recognition. In Proceedings of the 31st ACM International Conference on Multimedia (pp. 5923-5934).

**Suitability:**

3

---

### Official Review · Reviewer_qMww · 2024-05-29

**Rating:** 5
**Confidence:** 3

**Summary:**

The paper proposes DQ-Former, a multimodal fusion framework to perform Emotion Recogntion in Conversations (ERC). DQ-Former utilizes a middle-fusion approach for multimodal feature fusion incorporated with dynamic modality priority learning aimed to effectively fuse information across modalities. Audio and Text modalities are considered and experiments are conducted on the MELD and IEMOCAP datasets.

The effectiveness of the proposed middle-fusion, dynamic priority modality learning, and robustness of the proposed framework to missing modalities are studied.

**Strengths:**

The motivation/need for a middle-fusion technique to capture cross-modal information is well-grounded. The need to adaptively manage relative importance of modalities (towards tasks such as ERC) is well-explained. Identification that most existing methods treat MERC as general multimodal task is of importance.

Ablation studies are provided to validate each component in the proposed method including scenario where modalities are missing.

**Limitations:**

While identifying the need to account for dynamic modality priority for multimodal ERC models, the proposed method utilizes only textual and acoustic modalities from MELD and IEMOCAP datasets while 3 modalities (visual, text and acoustic) are available. The reason behind omitting a modality is requested from the authors. Experiments with all 3 modalities would help demonstrate the significance of the proposed work.

Section 3.4 discusses "Dynamic Modality Priority Learning". It is mentioned that for each instance (utterance), ConNet provides confidence estimates for each modality (m) at $i^\text{th}$-layer. An ablation study reporting the frequency of a modality (text or acoustic in this case) being dominant across utterances for MELD and IEMOCAP dataset can help identify patterns within the datasets if there is a dominating modality across most of the utterances (visual modality can be included in this study as well). This would help demonstrate the significance of the priority learning approach and its generalization ability across other multimodal tasks.

**Suitability:**

3

---

### Official Review · Reviewer_eqdQ · 2024-06-05

**Rating:** 5
**Confidence:** 3

**Summary:**

The paper proposed a novel middle-fusion mechanism for multimodal attention

**Strengths:**

- The paper has very sound and robust experiments and ablation studies
- The proposed technique exhibits competitive performance on the MELD and IEMOCAP datasets
- The paper is very well-written and easy to follow. Clear motivation is provided for each of the decisions made, and has intuitive explanations for how to arrive at each of the ideas

**Limitations:**

- The proposed fusion technique only works with two modalities
- There is not enough clarity on the novelty of the proposed mechanism - i.e. the authors should aim to provide more information to better differentiate the proposed technique from existing mlddle and late fusion attention mechanisms.

**Suitability:**

3

---

### Meta-Review · Area_Chair_PuAe · 2024-07-03

**Recommendation:** Accept (Poster)
**Confidence:** 5

**Metareview:**

Test-Time Adaptation for Multimodal Sentiment Analysis